# Assessing BeiDou-3 PPP-B2b with Signal-in-Space Ranging Error (SISRE) and Its Performances in Positioning and ZTD Estimation

**DOI:** 10.3390/s25216700

**Published:** 2025-11-02

**Authors:** Guangxing Wang, Fen Li, Wenhai Zhou, Guo Chen, Zhiyong Zhu, Xiaomin Jia, Qing An

**Affiliations:** 1Key Laboratory of Smart Earth, Beijing 100094, China; wanggx@cug.edu.cn (G.W.); zhuzhiyong02@nudt.edu.cn (Z.Z.); echos5@163.com (X.J.); 2School of Geography and Information Engineering, China University of Geosciences, Wuhan 430078, China; zhouwh_cug@163.com; 3GNSS Research Center, Wuhan University, Wuhan 430079, China; 4Beijing Institute of Tracking and Telecommunications Technology, Beijing 100095, China; 5College of Electronic Science, National University of Defense Technology, Changsha 410003, China; 6Beijing Satellite Navigation Center, Beijing 100094, China; 7School of Artificial Intelligence, Wuchang University of Technology, Wuhan 430223, China; 120160450@wut.edu.cn

**Keywords:** BDS, PPP-B2b, signal-in-space ranging error (SISRE), differential code bias (DCB), zenith total delay (ZTD)

## Abstract

The PPP-B2b service of BeiDou-3 enables real-time precise point positioning (RT-PPP) through correction information contained in B2b signals, circumventing the reliance on ground-based network infrastructures. This study comprehensively evaluates the accuracy of PPP-B2b correction parameters and their impact on positioning and tropospheric zenith total delay (ZTD) estimation. The PPP-B2b DCB products exhibit good consistency with the Chinese Academy of Sciences (CAS) reference, with average differences below 1.2 ns and standard deviations within 0.11 ns, indicating comparable performance to CAS products. For BDS-3 satellites, PPP-B2b achieves a radial orbit accuracy of 0.07 m and a clock standard deviation of 0.17 ns, outperforming the Centre National d’Études Spatiales (CNES) real-time products in both aspects. For GPS satellites, the corresponding accuracies are 0.06 m and 0.20 ns. Kinematic PPP experiments using combined GPS and BDS-3 observations yield horizontal and vertical accuracies of 4.3 cm and 2.8 cm, respectively, comparable to CNES results, while the BDS-3-only solution performs better than CNES but is still slightly inferior to the CODE. The ZTD estimation accuracy reaches 1.8 cm for GPS+BDS-3 combinations and 2.3 cm for BDS-3-only cases. Overall, PPP-B2b delivers centimeter-level performance in real-time positioning and ZTD estimation, demonstrating strong potential as an independent, space-based precise service, though further improvement is required for GPS-only applications.

## 1. Introduction

The third generation of BeiDou Navigation Satellite System (BDS-3) was officially inaugurated in July 2020. The BDS-3 constellation consists of three geostationary orbit (GEO) satellites, three inclined geosynchronous orbit (IGSO) satellites, and 24 medium Earth orbit (MEO) satellites. Wang et al. characterized the signal-in-space ranging error (SISRE) of BDS-3 and proposed preliminary integrity standards, providing a fundamental understanding of the system-level accuracy and reliability [1]. Besides basic service of navigation, a free precise point positioning (PPP) service among other characteristic capacities has become available for users in China and its neighboring regions [2,3,4]. The BDS-3 system provides a comprehensive suite of services, including Radio Navigation Satellite Service (RNSS), international search and rescue, and global short message communication, ensuring worldwide coverage. Within China and adjacent regions, BDS-3 further delivers advanced services such as PPP, satellite-based augmentation, ground-based augmentation, and regional short message communication, thereby enhancing positioning accuracy and service reliability. The B2b signal, which supports RNSSs, is transmitted by MEO and IGSO satellites, whereas the PPP-B2b signal is broadcast by GEO satellites. This signal delivers precise offset parameters for BDS-3 and GPS, including corrections of orbits and satellite clocks, as well as differential code biases (DCB), which are expected to enable real-time positioning with submeter-level accuracy even in areas without internet coverage [5].

Before PPP-B2b emerged, real-time precise point positioning (RT-PPP) had been a focal point of research and industrial application, as it offered the potential for achieving centimeter-level accuracy in real time, without dependence on base-station networks or delays in accessing precise orbit and clock products. Since 2013, users have been able to achieve high-precision RT-PPP based on real-time products from the International GNSS Service (IGS) real-time service (RTS) via the NTRIP protocol [6,7]. High precision and availability of orbits and satellite clocks provided by RTS had been demonstrated, and the precision of real-time products for GPS and Galileo can reach approximately 2 cm in terms of SISRE [8,9,10]. Elsobeiey and Al-Harbi employed the IGS GPS RTS products to recover precise orbits and clocks, and their experiments confirmed that the positioning accuracy was improved by 50% compared to that based on IGS ultra-rapid products [11]. In experiments with real-time static positioning, centimeter- and decimeter-level accuracies were achievable in horizontal and the vertical positions, respectively [12]. However, the vulnerability of internet-based real-time PPP technology is evident, particularly in areas with poor network coverage, such as polar regions, deserts, and oceans.

By broadcasting offset parameters through GEO satellites, the BDS PPP-B2b service effectively compensates for the limitations of internet-based RT-PPP services by the IGS or other commercial agencies. It aims at providing free, real-time, and reliable satellite navigation and positioning services for users within the region of 80–155° E and 5–55° N, regardless of network communication environments [5,13].

The researchers used marine measurement data to demonstrate that the BDS-3 PPP-B2b’s positioning accuracy meets the requirements for centimeter-level horizontal and decimeter-level vertical real-time high-precision positioning at sea [14]. With the PPP-B2b corrections applied, the orbit accuracies were improved by an average of 4.3%, 6.2%, and 16.1% in radial, normal, and tangential directions, respectively, while satellite clock errors were reduced from 1.18 ns to 0.22 ns [15]. PPP-B2b signals were collected using customized receivers at several stations in China, and centimeter- and decimeter-level accuracies were obtained in static and kinematic PPP experiments [16,17,18]. In addition, previous studies have shown that using an uncombined RT-PPP model with combined BDS-3/GPS observations, or integrating Galileo high accuracy service (HAS) with BDS-3 PPP-B2b, can further improve PPP-B2b positioning performance [19,20]. Meanwhile, its accuracy in zenith total delay (ZTD) and precipitable water vapor (PWV) retrieval has also been validated [21]. The performance of PPP-B2b with marine measurements was investigated to demonstrate that the requirements for centimeter-level horizontal and decimeter-level vertical positioning could be met in oceanic areas [14].

Building upon previous studies, this work further expands and refines both the research perspective and methodology. Specifically, based on the system-level characterization of BDS-3 SISRE by Wang et al., the SISRE metric is introduced into the accuracy evaluation of PPP-B2b correction products, with a systematic analysis of the impacts of orbit and clock errors on SISRE, enabling a more comprehensive performance characterization [1]. Meanwhile, an ionosphere-free PPP model is adopted to systematically compare PPP-B2b products with the final product from the Center for Orbit Determination in Europe (CODE) and the real-time product from the Centre National d’Études Spatiales (CNES), and validation is performed using data from the International GNSS Monitoring and Assessment System (iGMAS) stations to improve the engineering applicability of the evaluation results. Furthermore, unlike most existing studies that focus on a single application aspect (such as positioning accuracy or ZTD estimation), this study extends the evaluation dimension by jointly analyzing the PPP-B2b service performance in both high-precision positioning and tropospheric zenith total delay estimation, thereby verifying its potential for real-time atmospheric parameter retrieval and achieving a comprehensive assessment of PPP-B2b from precision to application.

The remainder of this paper is organized as follows. Section 2 presents the evaluation methodology and results for the accuracy of the PPP-B2b correction products, covering DCB, precise orbit and clock, and SISRE. Section 3 details the experimental setup and comparative results of PPP and ZTD estimation under different constellations and correction products. Section 4 summarizes the main findings and analyzes the performance of the PPP-B2b product in comparison with the CODE final product and the CNES real-time product, highlighting its potential advantages as well as existing limitations.

## 2. Methods

Along with correction parameters, a group of issue of data (IOD) is also broadcast through the PPP-B2b signal to ensure the inter-relationship among the information contents of different message types [13,22]. Since SISRE is regarded as a key performance indicator for satellite orbit and clock accuracy, it is used in this study to assess the performance of PPP-B2b correction information [23]. The computation of code biases, orbit and satellite clock with PPP-B2b corrections, as well as corresponding SISRE are introduced in this section.

### 2.1. Differential Bias Correction

Due to different satellite tracking modes, the offset contained in global navigation satellite system (GNSS) code measurement is different from channel to channel. When code measurements of more than one frequency are involved in data processing, the DCB should be taken into account as follows [13]:(1)$${\tilde{l}}_{\mathrm{sig}}=l_{sig}-D_{sig}$$
where ${\tilde{l}}_{sig}$ is the corrected code measurement, $l_{sig}$ is the code measurement directly captured by the receiver, and $D_{sig}$ is the DCB corresponding to the signal.

### 2.2. Precise Orbit Recovery

The satellite orbit correction parameters provided by the PPP-B2b signal can be expressed as(2)$$\delta O_{B2b}={\left[\begin{array}{ccc} \delta O_{R} & \delta O_{A} & \delta O_{C} \end{array}\right]}^{T}$$
where $\delta O_{R}$, $\delta O_{A}$, and $\delta O_{C}$ are the radial, tangential, and normal components of the orbit correction in the orbital coordinate system, respectively. Satellite position computed from the GPS navigation message (LNAV) or the BDS-3 B1C signal civil navigation message (CNAV1) is expressed in the Earth-centered Earth-fixed (ECEF) coordinate system, while the orbit correction broadcast through PPP-B2b is expressed as a vector with radial, along-track, and cross-track components. It is necessary to convert them to the ECEF coordinate system before calculating the precise satellite orbits with PPP-B2b correction. This method can be expressed as(3)$$X_{B2b}=X_{\mathrm{brdc}}-\left[\begin{array}{ccc} e_{R} & e_{A} & e_{C} \end{array}\right]\cdot \delta O_{B2b}$$
where $X_{B2b}$ is the satellite position corrected by the PPP-B2b signal, $X_{\mathrm{brdc}}$ is the satellite position obtained from the broadcast ephemeris, and $e_{R}$, $e_{A}$, and $e_{C}$ are the unit vectors in the radial, tangential, and normal directions of the satellite orbit in the ECEF coordinate system, respectively.

### 2.3. Precise CLOCK RECovery

The clock corrections contained in the PPP-B2b signal can be applied as(4)$$\Delta t=\Delta t_{\mathrm{brdc}}-\delta t_{B2b}$$
where Δt is the calculated precise satellite clock, $\Delta t_{\mathrm{brdc}}$ is the satellite clock retrieved from the CNAV1 navigation message of the B1C signal, and $\delta t_{B2b}$ is the satellite clock correction parameter extracted from the PPP-B2b signal. Note that the units of all three terms should be consistent, either expressed in length or in time.

### 2.4. SISRE Calculation

The SISRE characterizes the combined impact of orbit and satellite clock errors projected along the line-of-sight direction, and its magnitude therefore depends on the relative geometry between the satellite and the receiver. The average of SISREs over the Earth’s surface is usually assessed in terms of global-average SISRE and orbit-only SISRE, whose main difference lies in whether satellite clock error is taken into account [23,24]. Theoretically, global-average SISRE represents the root mean square error (RMS) of instantaneous SISREs all over the Earth’s surface, while orbit-only SISRE has the same meaning, except that only the orbit errors are involved [25,26]. In practice, the two types of SISRE are calculated as follows [23]:(5)$$SISRE_{\mathrm{ave}}=\sqrt{[\mathrm{rms}\left(w_{R}\delta_{R}-{c*\delta }_{\tau }\right){]}^{2}+\left(\sigma_{A}^{2}+\sigma_{C}^{2}\right)w_{\mathrm{AC}}^{2}}$$(6)$$SISRE_{\mathrm{orb}}=\sqrt{{\left(w_{R}\sigma_{R}\right)}^{2}+\left(\sigma_{A}^{2}+\sigma_{C}^{2}\right)w_{\mathrm{AC}}^{2}}$$
where $\delta_{R}$ and $\delta_{\tau }$ are the orbit error in radial direction and satellite clock error; c is the light velocity; $\sigma_{R}$, $\sigma_{A}$, and $\sigma_{C}$ are the RMS values of the radial, along-track, and cross-track projections of orbit error; and $w_{R}$ and $w_{\mathrm{AC}}$ are the weighting factors related to constellation and orbit type. The operator rms() represents the calculation of RMS.

## 3. Product Assessment

The performance of the PPP-B2b products is evaluated using state space representation (SSR) correction data collected over seven consecutive days from 20–26 August 2023. To establish a reference for accuracy assessment, we employed the final post-processed products from Wuhan University (WHU) and the DCB products released by the Chinese Academy of Sciences (CAS). The IGGDCB method, developed by the CAS, introduces innovative improvements to address the limitations of conventional DCB estimation techniques in regional monitoring networks. Its DCB products exhibit high stability, with the mean monthly stability better than 0.08 ns for GPS and 1.12 ns for BDS satellites. WHU, as the only international analysis center that provides orbit and clock products for all BDS satellites (including GEO, IGSO, and MEO), offers highly reliable reference data for orbit and clock accuracy assessment [27,28]. The final WHU products achieve orbit accuracies at the decimeter level for GEO and IGSO satellites and at the centimeter level for MEO satellites, with clock accuracy of approximately 0.1 ns [29,30]. Furthermore, to better evaluate the performance of PPP-B2b relative to other widely used products, a comparative analysis is conducted using the final products from CODE and the real-time products provided by CNES.

### 3.1. Code Bias Accuracy

Based on the DCB products by CAS on 22 August 2023, we evaluated the accuracies of DCBs provided through the PPP-B2b signal. The C1P-C6I and C2I-C6I DCBs of 27 BDS-3 satellites (specifically C19–C46, excluding C31, which is still under in-orbit testing) are shown in the upper and the lower panels of Figure 1, and for each satellite the DCB products of PPP-B2b and CAS are displayed as blue and orange dots. Although the DCBs are significantly different from satellite to satellite, the DCBs retrieved from the PPP-B2b signal are highly consistent with those provided by CAS. The absolute mean differences between the two kinds of DCB products, calculated over a 7-day period from 20–26 August 2023, as well as their corresponding standard deviations (STDs), are shown in Figure 2, and the colors of the bars are employed to distinguish DCBs with different signal pairs, i.e., blue for C1P-C6I and orange for C2I-C6I. With respect to the DCB products by CAS, the average differences for the C1P-C6I and C2I-C6I DCBs of the PPP-B2b products are 0.897 ns and 1.132 ns, respectively, and the corresponding standard deviations are 0.102 ns and 0.097 ns.

### 3.2. Orbit Comparison

Taking the final products from WHU as the reference, we assessed the orbit accuracies of GPS satellites (G02–G32, excluding G11/G14/G17/G18/G21/G28) and BDS-3 satellites (C19–C46, excluding C31) with PPP-B2b corrections and provided comparative analyses with the CODE and CNES RTS products.

As is shown in Table 1 and Figure 3, Figure 4 and Figure 5, the PPP-B2b orbit correction products demonstrate good performance in the radial direction, with RMS errors of 0.063 m and 0.068 m for the GPS and BDS-3 systems, respectively. The standard deviations in the radial direction are smaller than those in the along-track and cross-track directions, and the radial accuracy is comparable to that of the precise orbit products provided by CNES and CODE. Notably, for the BDS-3 system, the PPP-B2b slightly outperforms the CNES solution in terms of radial accuracy. In the along-track and cross-track directions, the orbit errors of PPP-B2b are still significantly larger than those of orbit products derived from global GNSS tracking networks. For example, if IGSO satellites (C38, C39, and C40) are excluded, the along-track orbit errors of CODE are generally within 0.05 m, and those of CNES are mostly within 0.2 m. The errors of PPP-B2b in both along-track and cross-track directions commonly exceed 0.2 m. This discrepancy primarily stems from differences in the underlying data sources. The products of CODE and CNES rely on globally and uniformly distributed GNSS tracking networks, which guarantee stronger observing geometry and greater data redundancy. In contrast, PPP-B2b is based mainly on a regional GNSS tracking network, leading to limitations in observing geometry [2].

When considering only IGSO satellites, both the CODE final products and the CNES RTS products exhibit larger errors than their MEO counterparts. In contrast, PPP-B2b exhibits superior radial accuracy for IGSO satellites, indicating a more effective orbit determination capability for these satellites. This advantage is likely due to PPP-B2b’s specific orbit correction strategies and its error modeling approach that is better adapted to the orbital characteristics of BDS-3 [31].

### 3.3. Clock Error

The second-order differencing method is used to evaluate the accuracy of precise clock products [2]. For each epoch, the satellite clock differences relative to the WHU final products are computed for each satellite to obtain first-order differences. The mean first-order difference across all satellites at the same epoch is then subtracted to derive the second-order differences.

Figure 6 and Figure 7 illustrate the second-order clock difference sequences for GPS and BDS-3 satellites from the PPP-B2b product on DOY 234. As shown, there is a clear similarity between the clock bias accuracy and the orbit error characteristics in PPP-B2b, with BDS-3 satellites exhibiting more stable and continuous clock accuracy compared to GPS. Nonetheless, both GPS and BDS-3 satellites show distinct color banding, indicating the presence of satellite-specific non-zero mean biases in the real-time PPP-B2b clock corrections. These systematic satellite-specific biases may stem from pseudo-range-related errors [32].

Figure 8 and Figure 9 present the weekly average RMS and standard deviation (STD) values of the second-order clock differences for GPS and BDS-3 satellites across different products. Detailed numerical values are provided in Table 2. Consistent with the single-day observations, the RMS of the second-order differences for BDS-3 in the PPP-B2b product is generally smaller than that for GPS, with a weekly mean RMS of 1.800 ns for BDS-3 and 3.005 ns for GPS. This reflects the limitation of the regional GNSS monitoring network for GPS and the benefit of additional inter-satellite link (ISL) observations for BDS-3. Additionally, the average RMS for both BDS-3 and GPS is significantly larger than their corresponding STD values, further indicating the presence of systematic biases in the PPP-B2b clock corrections. Since such biases are absorbed into range residuals and ambiguities during PPP processing; they do not impact the final positioning accuracy after convergence. Therefore, STD is the more appropriate indicator for evaluating the accuracy of PPP-B2b clock corrections [33]. In terms of average clock accuracy for GPS satellites, the PPP-B2b product performs slightly worse than CNES (0.171 ns) and CODE (0.139 ns). For BDS-3, however, the average clock STD from PPP-B2b (0.171 ns) is better than that of CNES RTS (0.620 ns) and CODE (0.213 ns). The significantly poorer performance of CNES for BDS-3 clock products compared to GPS is likely due to the limited tracking coverage of BDS-3 by Multi-GNSS Experiment (MGEX) stations. This highlights the potential of PPP-B2b in real-time applications for BDS-3 satellites. In contrast, the final clock products from CODE do not exhibit such disparity between GPS and BDS-3, with average RMS values of 0.505 ns for GPS and 0.594 ns for BDS-3.

Figure 8 and Figure 9 illustrate that the clock sequences for BDS-3 satellites in the PPP-B2b precise products are more stable and continuous. The significant difference in clock bias accuracy between GPS and BDS-3 in PPP-B2b may be due to the incompleteness of the regional GNSS network, which makes it difficult to accurately separate initial clock bias from ambiguities. However, the additional observations from the ISL terminals on BDS-3 satellites are beneficial to both orbit determination and clock estimation.

Due to the limited distribution of MGEX BDS-3 tracking stations, the average clock RMS for CNES GPS satellites is 0.556 ns, while the average clock error for BDS-3 satellites is 0.821 ns. The CODE final products do not exhibit such a difference in clock accuracy between GPS and BDS-3. The average clock RMS for GPS in the CODE final products is 0.505 ns, and for BDS-3 it is 0.594 ns. There are satellite-specific system biases in the real-time clock offsets of PPP-B2b. These satellite-specific system biases might be caused by pseudo-range biases. Fortunately, the ambiguity parameters in PPP can absorb the system biases so they do not affect the positioning accuracy. Therefore, the STD of the precise clock offset errors in PPP-B2b is used to evaluate the accuracy of the PPP-B2b clock offset corrections. In summary, the PPP-B2b product demonstrates competitive, and for BDS-3, superior clock accuracy compared to the other products.

### 3.4. SISRE Assessment

This section calculates the global-average SISRE for the PPP-B2b, CODE final products, and CNES RTS products. To better understand the contributions of orbit and clock errors to global-average SISRE, this study also computes orbit-only SISRE, with the statistical results shown in Table 3.

Figure 10 presents a comparative analysis of global-average SISRE, orbit-only SISRE, and clock errors for each product. It is evident from the figure that global-average SISRE is primarily influenced by clock errors—satellites with larger clock errors also exhibit higher global-average SISRE values. The RMS and STD indicators further reveal that the RMS is consistently much larger than the STD, indicating the presence of systematic biases in SISRE as well.

Focusing on the STD of global-average SISRE, which is more indicative of the final PPP accuracy, the PPP-B2b GPS product shows an average accuracy of 0.208 m, which is significantly worse than that of the CNES product (0.037 m). This aligns with PPP-B2b’s relatively lower orbit and clock accuracy for GPS satellites. In contrast, the STD of the PPP-B2b BDS-3 product (0.083 m) outperforms that of the CNES RTS product (0.131 m). CODE maintains the highest level of accuracy across both GPS and BDS-3 systems.

## 4. Practical Application

Experiments of PPP were conducted with observables collected at stations jfng, chu1, gua1, lha1, sha1, and xia1 from 20–26 August 2023, with a sampling interval of 30 s, and the distribution of the stations is shown in Figure 11. Performance of positioning as well as ZTD estimation using PPP-B2b corrections are investigated with GPS-only, BDS-3-only, and GPS+BDS-3 observables, respectively, and results using the final product WHU are taken as control.

Following the release of ITRF2020 in April 2022, a new IGS Reference Framework (IGS20) and a revised IGS antenna calibration document (IGS20.atx) were developed accordingly. IGS announced its transition from the IGS14/IGS14.atx framework to the new IGS20/IGS20.atx framework effective on 27 November 2022 (IGS 2023). Given the potential impact of updating to the new reference frames IGS20 and IGS20.atx on precision single point positioning results, it is essential to compare the PPP results using both IGS14.atx and IGS20.atx in this study to determine the most suitable version of the antenna calibration file for final application analysis regarding PPP-B2b products and other related products.

From Figure 12, when computing the kinematic PPP solution, there was no significant difference in the results of the GPS+BDS-3 satellite solution using IGS20.atx or IGS14.atx. The average horizontal accuracy was 3.56 cm and 3.58 cm, respectively, with almost identical elevation accuracy of approximately 5.06 cm. However, under static PPP solution, slight differences were observed in the results of the GPS+BDS-3 satellite solution using IGS20.atx or IGS14.atx, with average horizontal accuracies of 2.07 cm and 2.89 cm, and elevation accuracies of 1.37 cm and 1.13 cm, respectively. In conclusion, the choice between IGS20.atx and IGS14.atx for positioning applications yields negligible differences at the millimeter level. Consequently, the updated IGS20.atx was ultimately selected for participation in PPP-B2b product positioning and ZTD application processing, with comparisons made against CODE final products and CNES RTS products. The specific processing strategies are detailed in Table 4.

### 4.1. PPP Evaluation

Figure 13, Figure 14 and Figure 15 illustrate the kinematic PPP solution sequences for station JFNG on 22 August 2023, using three types of correction products: PPP-B2b, CODE final products, and CNES RTS products (where G denotes GPS-only, C denotes BDS-only, and GC denotes the combined GPS/BDS system). As shown in the figures, the GPS+BDS-3 combined solutions achieved the highest positioning accuracy across all products, with CODE providing the best overall performance, while PPP-B2b and CNES exhibited comparable results. When examining the GPS and BDS-3 satellite systems separately, the PPP-B2b BDS-3-only solution achieved RMS errors of 0.026/0.016/0.062 m in the East (E), North (N), and Up (U) directions, which is significantly better than the GPS-only solution (0.056/0.046/0.154 m). A direct comparison of BDS-3-only solutions reveals that PPP-B2b outperforms CNES, with improvements of 45%, 16%, and 11% in the E, N, and U components, respectively, highlighting the advantage of PPP-B2b in supporting BDS-3 satellites.

Figure 16 further presents the variation in the number of available satellites and position dilution of precision (PDOP) values during the same PPP processing, based on the PPP-B2b products. The average number of matched SSR satellites for GPS-only was 5.8, approximately 32% fewer than the 8.5 satellites for BDS-3, which explains the discontinuities and larger errors observed in the GPS-only solution. Moreover, the PDOP values during the BDS-3-only solution were consistently better than those of the GPS-only case, indicating that the geometric configuration of the BDS-3 constellation is more favorable for achieving higher positioning accuracy.

Figure 17 and Figure 18 display the statistical results of positioning accuracy for kinematic PPP and static PPP at six stations over seven consecutive days, respectively, reflecting the average RMS values in the E, N, and U directions and the number of available satellites under three observation schemes: GPS-only, BDS-only, and GPS+BDS-3. From the kinematic PPP results, the PPP-B2b products using GPS+BDS-3 observations achieve centimeter-level accuracy in both horizontal (4.3 cm) and vertical (2.8 cm) positioning across all stations. When using only BDS-3 satellites, the PPP-B2b products exhibit better horizontal (5.6 cm) and vertical (7.3 cm) positioning accuracy than the CNES RTS products (6.7 cm/7.7 cm). However, Figure 17 clearly shows that when using only GPS satellites, the PPP-B2b positioning accuracy performs the worst among the three schemes. For static PPP, all three products achieve centimeter-level accuracy with either GPS-only or BDS-3-only observations, with the best accuracy obtained when using GPS+BDS-3 observations across all stations. When using only BDS-3 satellites, PPP-B2b still demonstrates better performance than CNES.

Overall, the positioning performance of the PPP-B2b products using the GPS+BDS-3 system is comparable to that of the CNES RTS products. When using only the BDS-3 system, PPP-B2b slightly outperforms CNES RTS, but when using only the GPS system, the positioning performance of PPP-B2b still needs improvement.

### 4.2. ZTD Evaluation

This section aims to evaluate the performance of the PPP-B2b product in estimating the ZTD. The ZTD estimates obtained from three positioning strategies—GPS-only, BDS-3-only, and GPS+BDS-3—are compared against reference values provided by iGMAS, whose final ZTD products have an accuracy better than 4 mm with a sampling interval of 2 h [34]. For benchmarking purposes, the results from the CODE final product and the CNES RTS product are also included as comparative references.

Figure 19 presents the ZTD estimation errors under both kinematic and static positioning modes for six selected stations. Overall, the ZTD estimation accuracy of the PPP-B2b product exhibits minimal differences between the static and kinematic modes, indicating a good level of temporal and spatial stability. However, significant variations are observed across the different positioning strategies. Among them, the GPS-only solution of PPP-B2b shows the largest estimation error, with accuracy maintained within 3 cm. The BDS-3-only solution yields an error of 2.3 cm, while the GPS+BDS-3 solution achieves the highest accuracy at 1.8 cm. These results suggest that the PPP-B2b product performs better in BDS-3-only and multi-constellation configurations with respect to ZTD estimation.

Table 5 summarizes the maximum RMS errors of ZTD estimates for each product under different constellation configurations. The CODE final product demonstrates the best performance, with ZTD errors below 1.5 cm for GPS-only, BDS-3-only, and GPS+BDS-3 solutions. The CNES RTS product also shows good performance, with ZTD errors within 1.7 cm for all three configurations, consistently outperforming PPP-B2b.

In summary, the evaluation results indicate that the ZTD estimation accuracy of the PPP-B2b product under the BDS-3-only solution is superior to that under the GPS-only solution, highlighting its potential and advantages under BDS-3 support. Nevertheless, in comparison to the CODE final product and the CNES RTS product, the PPP-B2b product still lags in ZTD estimation accuracy—particularly under the GPS-only mode—suggesting that further improvements are needed in terms of correction quality and product stability.

## 5. Conclusions

As a space-based real-time correction service, BeiDou-3 PPP-B2b effectively addresses the network-dependent constraints of IGS real-time products, thereby providing an independent and efficient solution for real-time, high-precision positioning. This study systematically evaluated the accuracy of PPP-B2b correction parameters, the positioning performance of PPP, and the impact on ZTD estimation, with comparisons made against the CODE final product and the CNES real-time product.

The consistency of PPP-B2b DCB products was assessed using the MGEX DCB products provided by the CAS as a reference. The average differences in the C1P–C6I and C2I–C6I DCBs for the BDS-3 constellation in the PPP-B2b product were 0.897 ns and 1.132 ns, respectively, with STDs below 0.11 ns. These results demonstrate that the DCB performance of PPP-B2b is comparable to that of the CAS product.

Subsequently, using the final products from WHU as reference, the satellite orbit and clock accuracy of PPP-B2b, CODE, and CNES were evaluated, and the SISRE and its orbital components were computed. The results reveal that all three products achieved centimeter-level radial orbit accuracy for both GPS and BDS-3 satellites, confirming the effectiveness of PPP-B2b corrections in precise orbit determination. In terms of average clock accuracy for GPS satellites, PPP-B2b was slightly inferior to CNES (0.171 ns) and lower than CODE (0.139 ns). For BDS-3 satellites, the average clock accuracy of PPP-B2b (0.171 ns) outperformed both CNES RTS (0.620 ns) and CODE (0.213 ns), which can be attributed to the limited coverage of regional GNSS monitoring networks and the additional observations from the ISL terminals onboard BDS-3 satellites. Analysis of orbit and clock errors indicates that the SISRE of PPP-B2b is primarily determined by clock errors, with the BDS-3 satellite SISRE STD (0.062 m) slightly better than that of CNES (0.074 m).

To further assess its performance in positioning and ZTD estimation, PPP experiments and ZTD estimation tests were conducted using GPS-only, BDS-3-only, and combined GPS+BDS-3 observations. The results indicate that the PPP-B2b service achieved centimeter-level positioning accuracy in the kinematic GPS+BDS-3 combined mode, comparable to the CNES real-time product. In the BDS-3-only scenario, PPP-B2b outperformed CNES, while the GPS-only solution of PPP-B2b still requires improvement. Static PPP results also demonstrated favorable performance for PPP-B2b. In terms of ZTD estimation, PPP-B2b exhibited better accuracy under the BDS-3-only scheme than the GPS-only scheme; however, its overall performance was inferior to that of the CODE final product and the CNES real-time product. Overall, the PPP-B2b correction service demonstrates clear advantages in real-time correction capability for the BDS-3 system, exhibiting high reliability and practical applicability.

Future development of the PPP-B2b service should focus on improving its global coverage and internal consistency. Expanding the spatial distribution of BDS monitoring stations, particularly in regions beyond East Asia, would significantly enhance the accuracy of orbit and clock corrections. The integration of multi-GNSS observations and the refinement of inter-system bias calibration can further strengthen the precision and robustness of the correction products. In addition, advanced tropospheric and ionospheric modeling based on machine learning techniques is expected to substantially improve the accuracy of real-time atmospheric delay estimation. Furthermore, the integration of PPP-B2b with low Earth orbit (LEO) satellite constellations and regional augmentation systems could significantly accelerate convergence, enhance service continuity and reliability, and provide stronger support for future PPP applications.

## Figures and Tables

**Figure 1 sensors-25-06700-f001:**
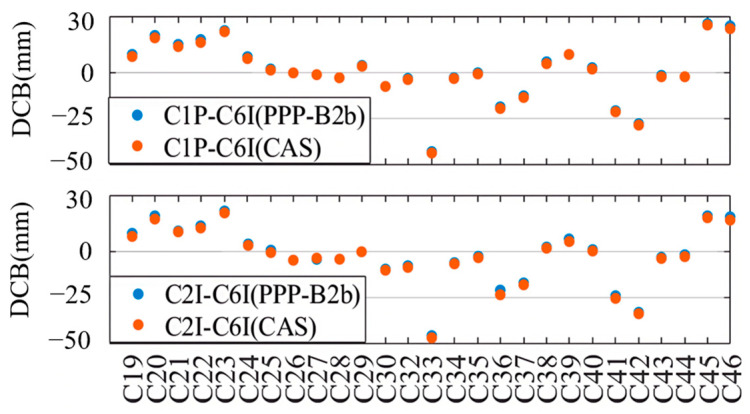
The DCB values of C1P-C6I (**upper**) and C2I-C6I (**lower**) for PPP-B2b and CAS.

**Figure 2 sensors-25-06700-f002:**
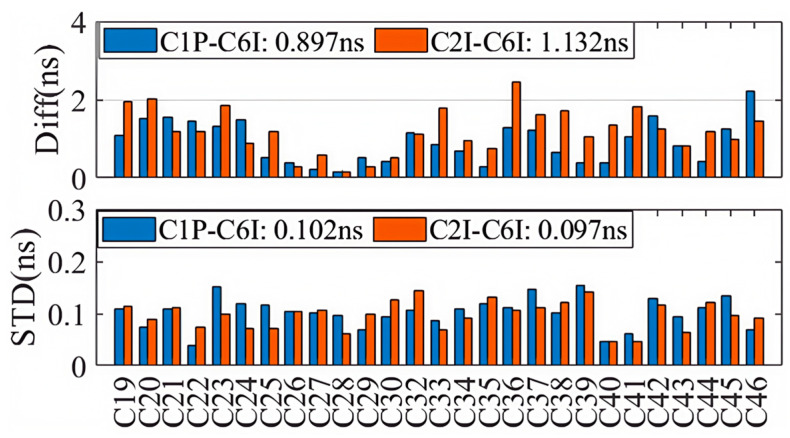
The difference and STDs between PPP-B2b DCB and CAS DCB.

**Figure 3 sensors-25-06700-f003:**
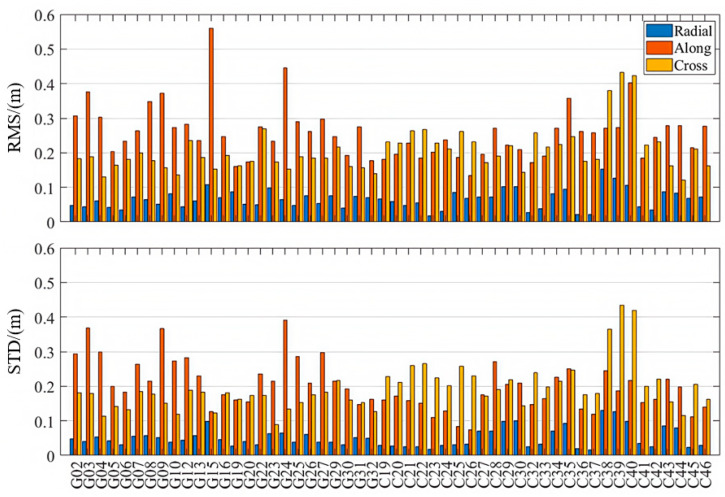
Orbit accuracies in radial, along-track, and cross-track directions (PPP-B2b).

**Figure 4 sensors-25-06700-f004:**
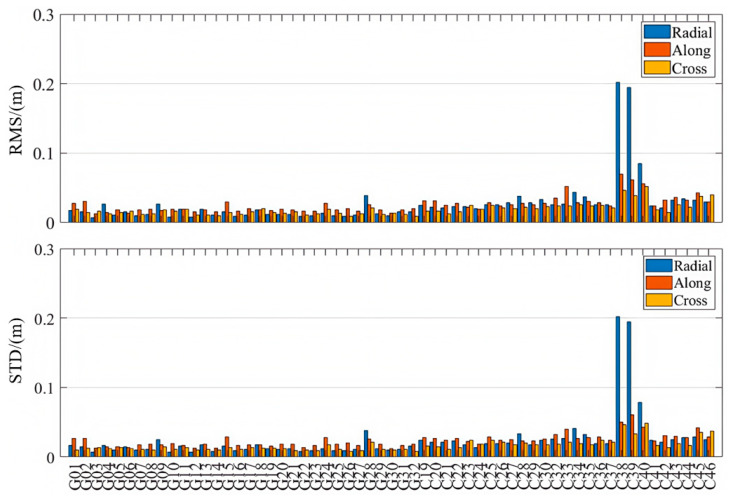
Orbit accuracies in radial, along-track, and cross-track directions (CODE).

**Figure 5 sensors-25-06700-f005:**
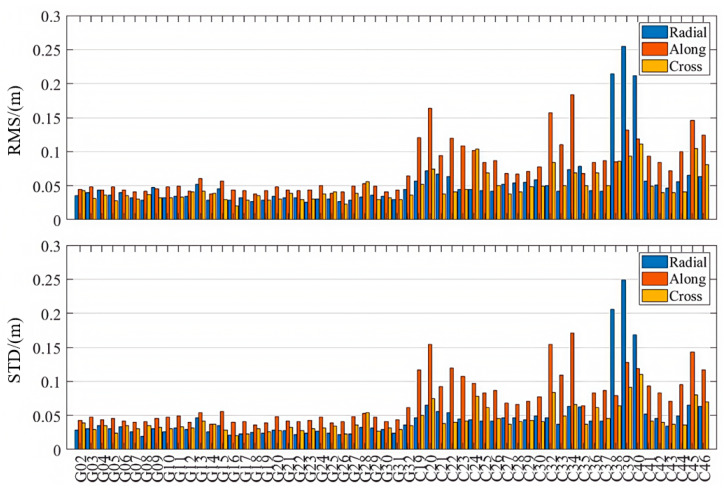
Orbit accuracies in radial, along-track, and cross-track directions (CNES).

**Figure 6 sensors-25-06700-f006:**
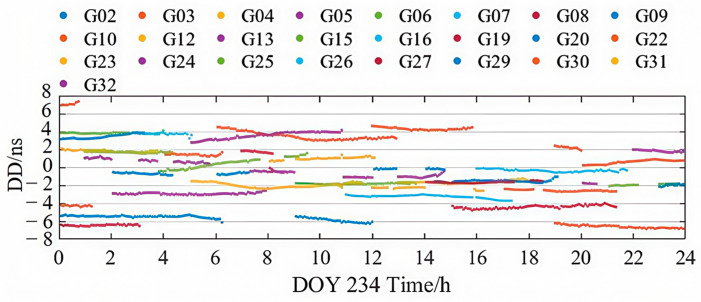
Time series of PPP-B2b GPS satellites clock errors on DOY 234, 2023.

**Figure 7 sensors-25-06700-f007:**
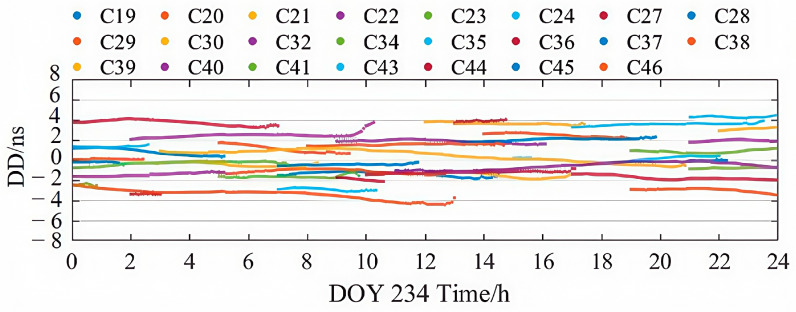
Time series of PPP-B2b BDS-3 satellites clock errors on DOY 234, 2023.

**Figure 8 sensors-25-06700-f008:**
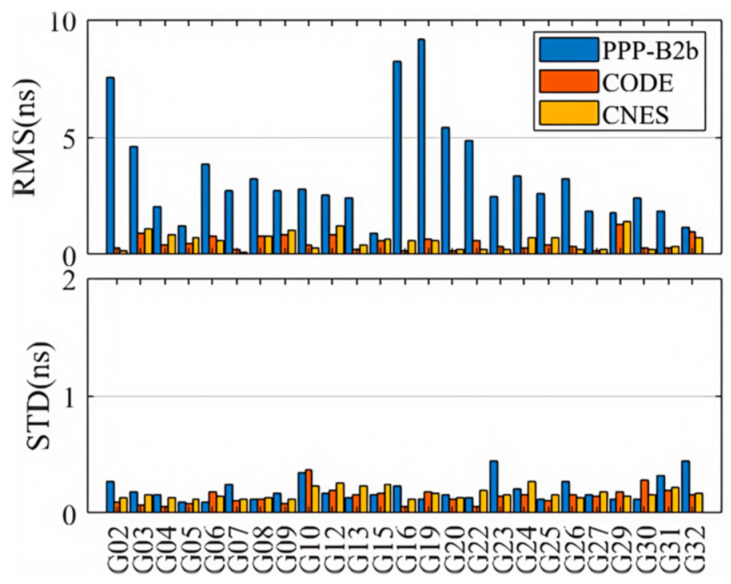
Clock accuracies of GPS satellites for PPP-B2b, CODE, and CNES products.

**Figure 9 sensors-25-06700-f009:**
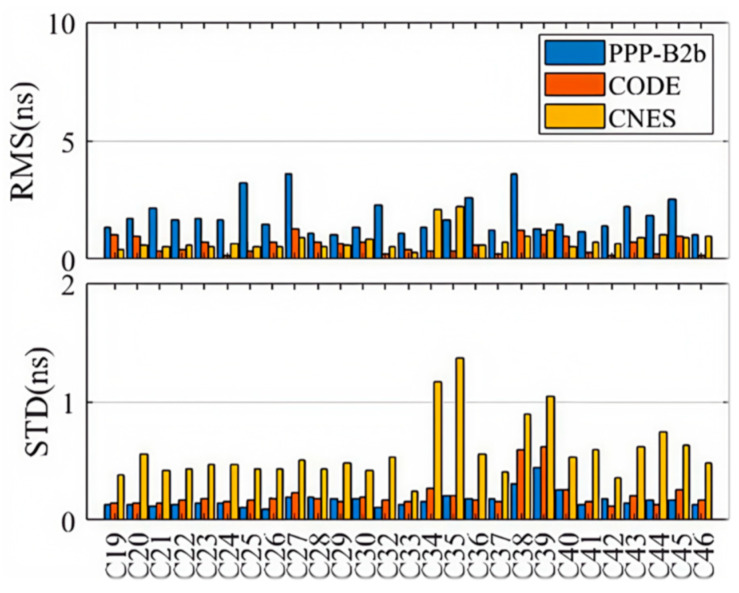
Clock accuracies of BDS-3 satellites for PPP-B2b, CODE, and CNES products.

**Figure 10 sensors-25-06700-f010:**
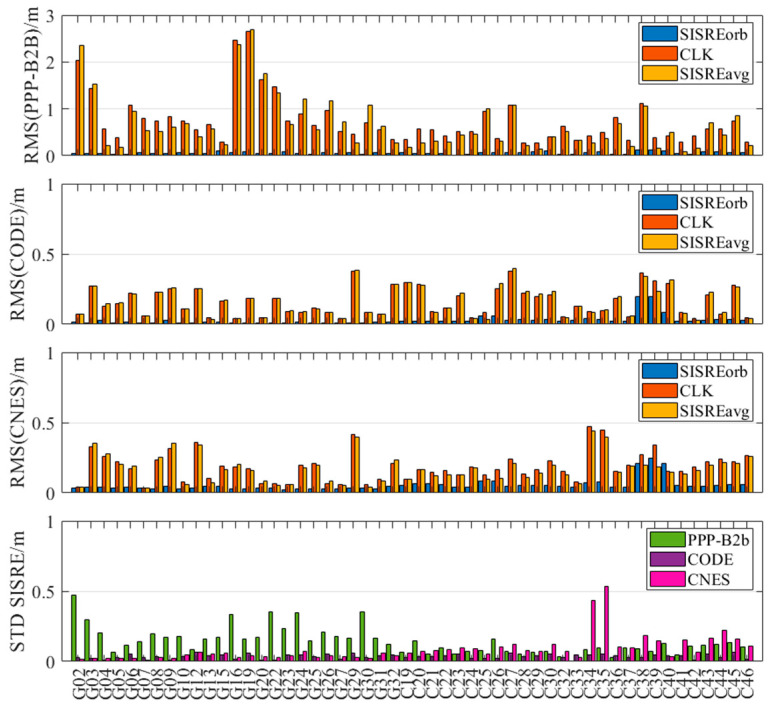
Comparison of $SISRE_{\mathrm{ave}}$, $SISRE_{\mathrm{orb}}$, and clock bias for each product.

**Figure 11 sensors-25-06700-f011:**
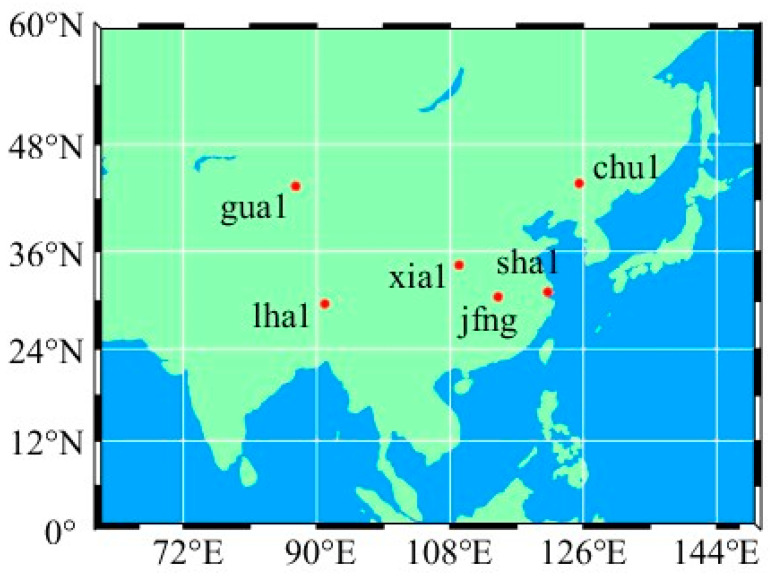
Distribution of the six stations.

**Figure 12 sensors-25-06700-f012:**
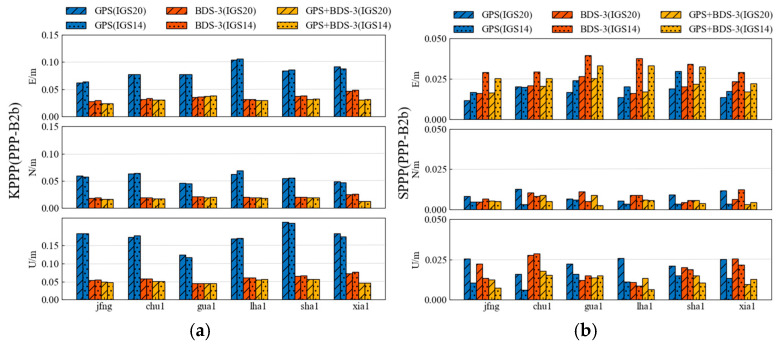
Weekly average results for kinematic (**a**) and static (**b**) PPP solutions utilizing IGS20.atx and IGS14.atx.

**Figure 13 sensors-25-06700-f013:**
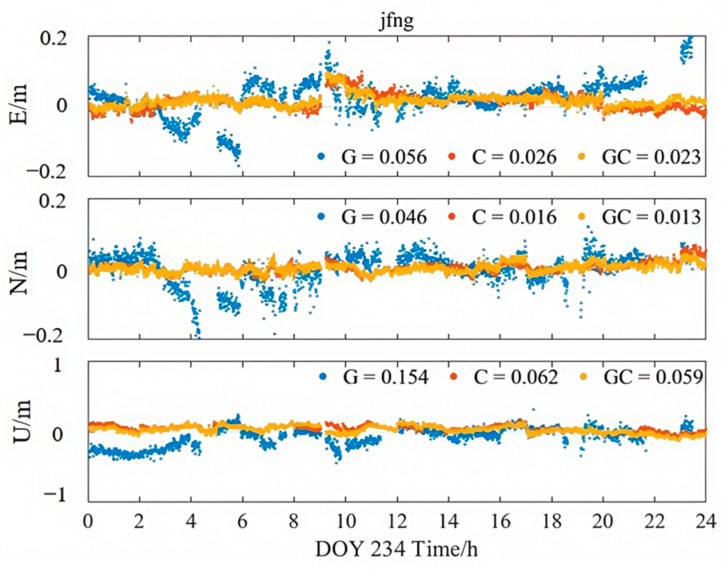
Kinematic PPP solution applying PPP-B2b products.

**Figure 14 sensors-25-06700-f014:**
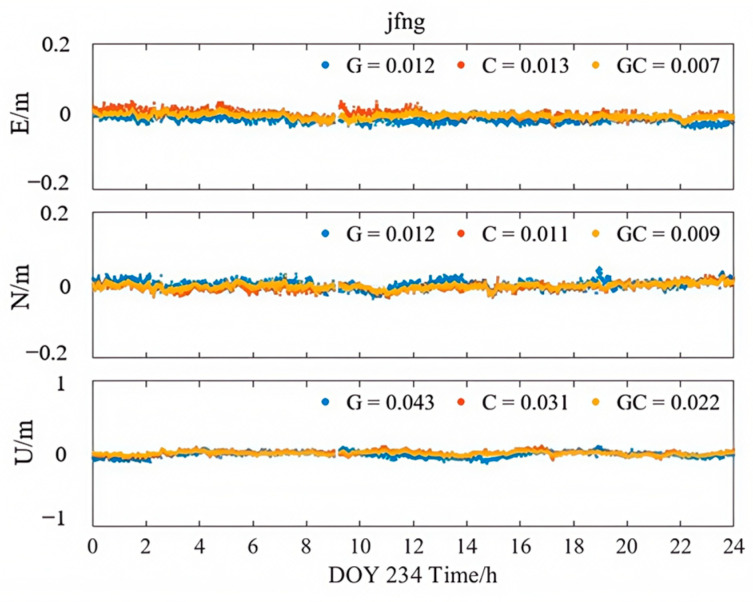
Kinematic PPP solution applying CODE products.

**Figure 15 sensors-25-06700-f015:**
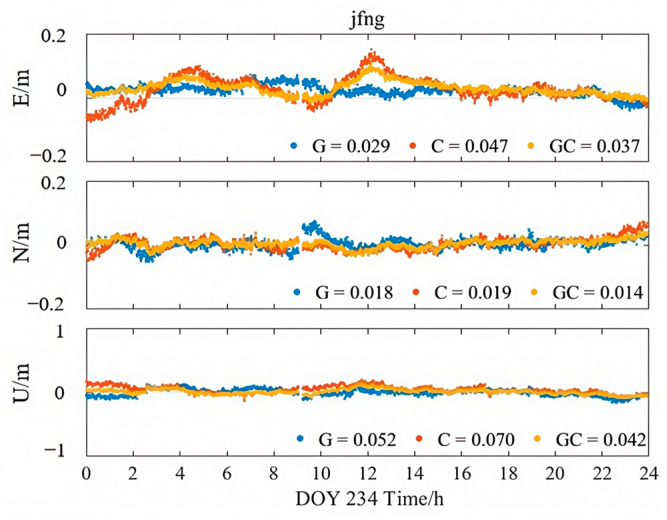
Kinematic PPP solution applying CNES products.

**Figure 16 sensors-25-06700-f016:**
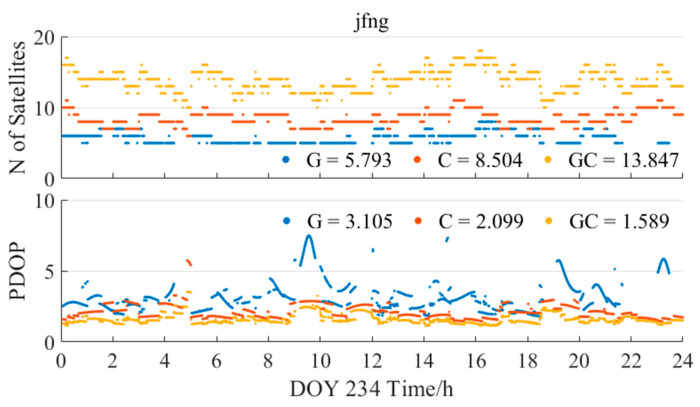
The number of available satellites and PDOP applying PPP-B2b products on the right.

**Figure 17 sensors-25-06700-f017:**
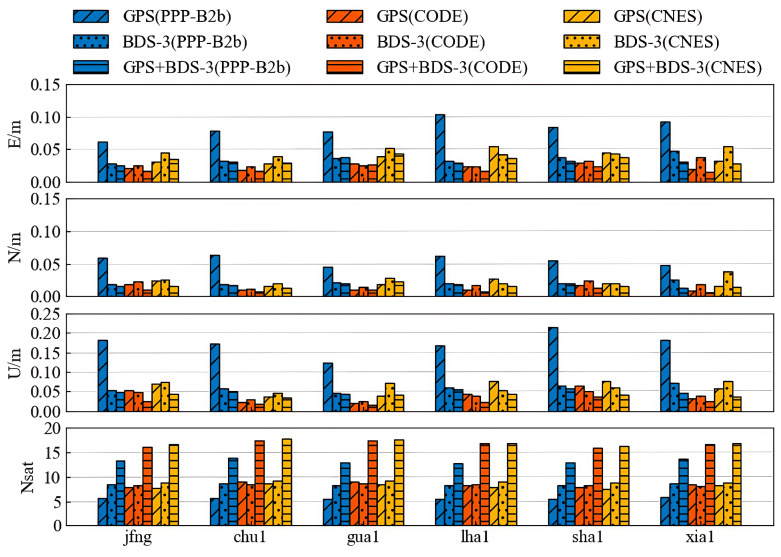
Weekly average results for kinematic PPP solutions.

**Figure 18 sensors-25-06700-f018:**
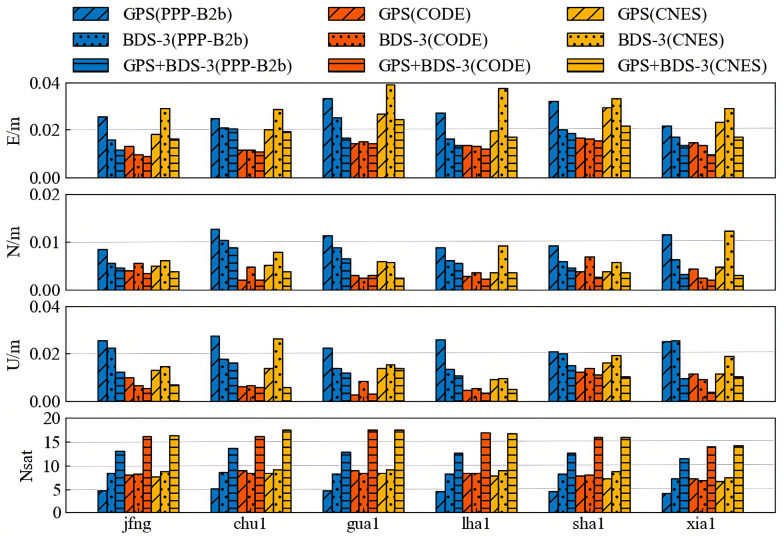
Weekly average results for static PPP solutions.

**Figure 19 sensors-25-06700-f019:**
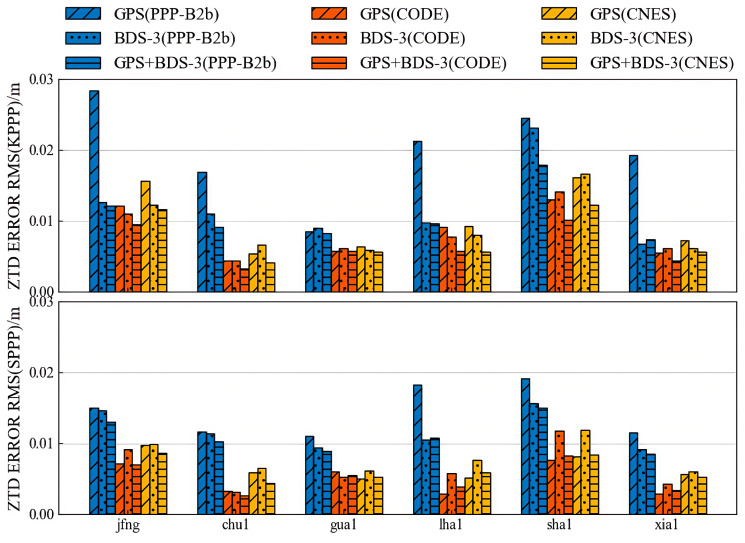
Weekly accuracies of ZTDs with GPS, BDS-3, and both.

**Table 1 sensors-25-06700-t001:** Average RMS statistics of orbit errors for each product.

System	PPP-B2b			CODE			CNES		
	Radial	Along-Track	Cross-Track	Radial	Along-Track	Cross-Track	Radial	Along-Track	Cross-Track
GPS	0.063	0.281	0.178	0.014	0.019	0.014	0.034	0.046	0.034
BDS-3	0.068	0.236	0.233	0.044	0.034	0.025	0.076	0.105	0.062

**Table 2 sensors-25-06700-t002:** Statistical comparison of the second-order clock differences for each product.

	GPS			BDS-3		
	PPP-B2b	CODE	CNES	PPP-B2b	CODE	CNES
RMS/ns	3.005	0.505	0.556	1.800	0.594	0.821
STD/ns	0.200	0.139	0.171	0.171	0.213	0.620

**Table 3 sensors-25-06700-t003:** Statistical comparison of SISRE for each product.

	GPS			BDS-3		
	PPP-B2b	CODE	CNES	PPP-B2b	CODE	CNES
$SISRE_{\mathrm{ave}}$ (RMS/m)	0.938	0.147	0.160	0.432	0.153	0.176
$SISRE_{\mathrm{orb}}$ (RMS/m)	0.055	0.013	0.033	0.062	0.044	0.074
$SISRE_{\mathrm{ave}}$ (STD/m)	0.208	0.035	0.037	0.083	0.038	0.131

**Table 4 sensors-25-06700-t004:** Processing strategies.

Content	Strategy
Observation type	IF: B1/B3, L1/L2
Interval	30 s
Cut-off elevation	5°
Satellite orbit and clock	PPP-B2b/CODE/CNES
PCO and PCV	IGS20.atx
Solid tide, ocean tide	IERS Conventions 2010
ZTD parameters	Saastamoinen model and random walk
Noise levels of code and phase	0.3 m/0.01 cycle
Ambiguity	Float

**Table 5 sensors-25-06700-t005:** Maximum RMS of estimated ZTD errors using each product.

Estimation Method		Maximum RMS of ZTD (cm)		
		PPP-B2b	CODE	CNES
Kinematic solution	GPS	3.0	1.3	1.6
	BDS-3	2.3	1.4	1.6
	GPS+BDS-3	1.8	1.0	1.2
Static solution	GPS	1.9	0.7	1.0
	BDS-3	1.5	0.9	1.2
	GPS+BDS-3	1.5	0.7	0.8

## Data Availability

The IGS MGEX observations can be downloaded from ftp://gdc.cddis.eosdis.nasa.gov (accessed on 16 October 2025). The GFZ final product can be downloaded from ftp://igs.gnsswhu.cn/ (accessed on 16 October 2025). The final product from CODE is available at: https://cddis.nasa.gov/archive/gnss/products/mgex/2198/ (accessed on 16 October 2025). The real-time product from CNES is available at: http://www.ppp-wizard.net/products/ (accessed on 16 October 2025).

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
