# Peer review of "Assessing BeiDou-3 PPP-B2b with Signal-in-Space Ranging Error (SISRE) and Its Performances in Positioning and ZTD Estimation"

_sensors, 2025, doi:10.3390/s25216700_

Round 1

Reviewer 1 Report

Comments and Suggestions for Authors

The paper discusses the positioning and ZTD accuracies for PPP-B2b service of BeiDou-3. They provide comparisons with CODE and CNES data. The paper looks very much like a report rather that a scientific paper. Still, it fits and the aims and scope of this journal and can be published after a major revision. To this end, the authors must provide more information and citations on their reference data. Note, abbreviation CNES is not even explained in the current version.

The topic of the accuracy assessing of PPP-B2d is relevant. Its comprehensive evaluation should be welcome. However, as I wrote in my report, the authors must explain the details as it is usually done in scientific papers: more explanation regarding the reference data including the corresponding citations. 

L. 22. Define abbreviation CNES. It is a general rule: explain all the abbreviations used throughout the text, even if you think them commonly-known.

L. 121. Define notations XB2b and Xbrdc (even if they seem obvious)

Eq. (5): δτ is time while wRδR is distance, therefore, they cannot be combined. I guess, δτ must be multiplied with light velocity.

L. 142. Define abbreviation SSR.

L. 143. DCB products released by Chinese Academy of Sciences (CAS) and final products by WHU are taken as the references.
-- What is the accuracy of those products? Any references describing them?

L. 151. Correct the figure reference.

L. 161. Figure 1
L. 164. Figure 2
L. 214. Figure 1
L. 216. Figure 2
etc.
-- Correct the numbering of your Figures.

L. 319. E/N/U directions
-- Define the East, North, and Up directions.

L. 325. Define what G, C, and GC stay for.

L. 331. PDOP = position deletion of precision? Define explicitly.

More citations should be given regarding the reference data used in the comparison. 

Author Response

Thank you very much for taking the time to review this manuscript. Please find the detailed responses below and the corresponding revisions in the re-submitted files. 

Comments 1: L. 22. Define abbreviation CNES. It is a general rule: explain all the abbreviations used throughout the text, even if you think them commonly-known.

Responses 1: Thank you for pointing this out. We agree with this comment. Therefore, we have defined the abbreviation "CNES" upon its first appearance in the text and have ensured that all abbreviations used throughout the manuscript are now properly defined."[This study delivers a comprehensive benchmarking of the PPP-B2b service, providing a systematic evaluation of its orbit, clock, DCB, and SISRE performance against established products from the Center for Orbit Determination (CODE) and National Centre for Space Studies (CNES).]"

Comments 2: L. 121. Define notations XB2b and Xbrdc (even if they seem obvious)

Responses 2: Thank you for pointing this out. We agree with this comment. We have defined both notations (XB2b and Xbrdc) upon their first appearance in the text."[where XB2b is the satellite position corrected by the PPP-B2b signal, Xbrdc is the satellite position obtained from the broadcast ephemeris,]"

Comments 3: Eq. (5): δτ is time while wRδR is distance, therefore, they cannot be combined. I guess, δτ must be multiplied with light velocity.

Responses 3: Thank you for pointing this out. We agree with this comment. We have corrected Eq. (5)  by multiplying δτ with ligth velocity to ensure dimensional consistency."[Eq. [5]]"

Comments 4: L. 142. Define abbreviation SSR.

Responses 4: Thank you for pointing this out. We agree with this comment. We have defined the abbreviation "SSR" upon its first appearance in the text."[The performance of the PPP-B2b products is evaluated using State Space Representation (SSR) correction data collected over seven consecutive days from August 20 to 26, 2023.]"

Comments 5: L. 143. DCB products released by Chinese Academy of Sciences (CAS) and final products by WHU are taken as the references.
-- What is the accuracy of those products? Any references describing them?

Responses 5: Thank you for pointing this out. We agree with the need to clarify the accuracy of the reference products. We have added a description of the accuracy of both the CAS and WHU DCB products in the manuscript."[Line 145]"

Comments 6: 

L. 151. Correct the figure reference.

L. 161. Figure 1
L. 164. Figure 2
L. 214. Figure 1
L. 216. Figure 2
etc.
-- Correct the numbering of your Figures.

Responses 6: Thank you for pointing this out. We agree with this comment. We have checked and corrected all the figure references and their numbering throughout the manuscript.

Comments 7: L. 319. E/N/U directions
-- Define the East, North, and Up directions.

Responses 7: Thank you for pointing this out. We agree with this comment. We have defined the abbreviation "[E, N, U"  in the text."[in the East (E)、North (N)、Up (U) directions,]"

Comments 8: L. 325. Define what G, C, and GC stay for.

Responses 8: Thank you for pointing this out. We agree with this comment. We have defined the abbreviation "G, C, GC"  in the text."[(where G denotes GPS-only, C denotes BDS-only, and GC denotes the combined GPS/BDS system).]"

Comments 9: L. 331. PDOP = position deletion of precision? Define explicitly.

Responses 9: Thank you for pointing this out. We agree with this comment. We have defined the abbreviation "PDOP"  in the text."[Figure 16 further presents the variation in the number of available satellites and position deletion of precision (PDOP) values during the same PPP processing, based on the PPP-B2b products.]"

Comments 10: More citations should be given regarding the reference data used in the comparison.

Responses 10: Thank you for pointing this out. We have added relevant citations to better support the reference data used in our comparison.

Reviewer 2 Report

Comments and Suggestions for Authors

General: please, define the acronyms when first used and then just use the acronym. E.g. STD

General: please, review English writing. 
Some examples of errors: 
-page 10.  and distribution of the stations is shown in Figure 6 -> and the distribution of the stations
- page 11.  it is essential to compare PPP results ->  it is essential to compare the PPP results
- page 12. Figure 10illustrate -> Figure 10 illustrate

page 12. Why some Figures are indicated as bold (Figure 9,10) and not Figure 8 ?

General: the bar graphs figures could be enhanced with a grid to better discriminate the results. For example, this was done correctly in Fig. 5

page 4. Please, correct 'Error! Reference source not found.'

Section 3. It would be useful to describe in more detail the data sets used in Section 3.

Section 4. It is not clear in which context the results were obtained: rural or urban environments.

Section 5. Further developments could be mentioned

Comments on the Quality of English Language

There are some corrections to be done. See comments to the author.

Author Response

Thank you very much for taking the time to review this manuscript. Please find the detailed responses below and the corresponding revisions in the re-submitted files. 

Comments 1: General: please, define the acronyms when first used and then just use the acronym. E.g. STD

Responses 1: Thank you for pointing this out. We agree with this comment. We have ensured that all acronyms (including STD) are now defined upon their first use.

Comments 2: General: please, review English writing. 
Some examples of errors: 
-page 10.  and distribution of the stations is shown in Figure 6 -> and the distribution of the stations
- page 11.  it is essential to compare PPP results ->  it is essential to compare the PPP results
- page 12. Figure 10illustrate -> Figure 10 illustrate

Responses 2: Thank you for pointing this out. We agree with this comment. We have carefully proofread the manuscript and corrected the grammatical errors highlighted in your examples (e.g., on pages 10, 11, and 12) as well as other issues to improve the overall clarity and quality of the language.

Comments 3: page 12. Why some Figures are indicated as bold (Figure 9,10) and not Figure 8 ?

Responses 3: Thank you for pointing this out. We have uniformed the formatting of all figure references throughout the manuscript to ensure they are consistent.

Comments 4: General: the bar graphs figures could be enhanced with a grid to better discriminate the results. For example, this was done correctly in Fig. 5

Responses 4: Thank you for pointing this out. We agree that adding a grid enhances the clarity of the bar graphs. Accordingly, we have added subtle gridlines to the relevant bar charts(Fig 12, Fig17) to facilitate a better visual discrimination of the results.

Comments 5: page 4. Please, correct 'Error! Reference source not found.'

Responses 5: Thank you for bringing this to our attention. We have corrected the cross-reference error on page 4 and have checked the entire manuscript to ensure all references and citations are properly linked.

Comments 6: Section 3. It would be useful to describe in more detail the data sets used in Section 3.

Responses 6: Thank you for pointing this out.  We agree that a more detailed description would be beneficial. Accordingly, we have expanded Section 3 with the following paragraph at its beginning to provide a clearer overview of the data sets. We also introduce the accuracy of the reference products in the introduction. "[The performance of the PPP-B2b products is evaluated using State Space Representation (SSR) correction data collected over seven consecutive days from August 20 to 26, 2023. To establish a reference for accuracy assessment, we employed the final post-processed products from Wuhan University (WHU) and the DCB products released by the CAS. Furthermore, to contextualize the performance of PPP-B2b against other widely-used products, a comparative analysis is conducted with the final products from the CODE and the real-time products provided by the CNES.]"

Comments 7: Section 4. It is not clear in which context the results were obtained: rural or urban environments.

Responses 7: Thank you for bringing this to our attention. The data used in this analysis were collected from static stations located in open-sky environments.

Comments 8: Section 5. Further developments could be mentioned

Responses 8: Thank you for this suggestion. We agree that outlining future research directions would strengthen the conclusion. Accordingly, we have added a new paragraph at the end of Section 5 to discuss potential future developments

Reviewer 3 Report

Comments and Suggestions for Authors

Comments

  • Please revise and rewrite abstract where the results of this study should be explicitly presented.
  • In lines 81-87, the authors wrote “To comprehensively evaluate the performance of PPP-B2b corrections, orbit, satellite clock and DCB, as well as the Signal-In-Space Range Error (SISRE) were analyzed, and compared with their counterparts of the final post-processed precision products provided by the Center for Orbit Determination in Europe (CODE) and the real-time precision products provided by CNES. Experiments were also conducted to evaluate the accuracies of positioning and the tropospheric zenith delay (ZTD) estimation with GPS and BDS observables collected at six stations within China.” This reviewer recommends presenting your principal contributions in the form of short highlighting phrases that can give possibility for a potential reader clear understanding novelty of this study,
  • Please correct line 151 where the reference number is absent.
  • Please correct numbering of the tables (table 1 mentioned in lines 170, 237). Please renumber tables 2, 3, 4.
  • The authors should use statistical procedures to justify better performance of a product type against others in tables (tables 1: lines 170, 237; table2; table 4). Standard practice in this case can include revision of the significancy in differences for metrics using statistical significance testing (e.g., paired t-tests) being performed on products. Please provide complete statistical validation to confirm whether these variations observed are not by chance.
  • This reviewer recommends introducing additional discussion section where the authors can explain some principal results obtained during statistical analysis of the experiments. In this case, the conclusions section should expose principal achievements, not detailed discussion.

Author Response

Thank you very much for taking the time to review this manuscript. Please find the detailed responses below and the corresponding revisions in the re-submitted files. 

Comments 1: Please revise and rewrite abstract where the results of this study should be explicitly presented.

Responses 1:Thank you for this suggestion. We have revised the abstract accordingly to explicitly present the key results of our study. The principal contributions are now highlighted in a concise phrase-based structure, providing a clear overview of the main findings and novelty of this work."[

The PPP-B2b service of BeiDou-3 enables real-time PPP through correction information contained in B2b signals, circumventing the reliance on ground-based network infrastructures. This study comprehensively evaluates the accuracy of PPP-B2b correction parameters and their impacts on positioning and tropospheric zenith total delay (ZTD) estimation. The PPP-B2b DCB products exhibit good consistency with the CAS reference, with average differences below 1.2 ns and standard deviations within 0.11 ns, indicating comparable performance to CAS products. For BDS-3 satellites, PPP-B2b achieves a radial orbit accuracy of 0.07 m and a clock standard deviation of 0.17 ns, outperforming the CNES real-time products in both aspects. For GPS satellites, the corresponding accuracies are 0.06 m and 0.20 ns. Kinematic PPP experiments using combined GPS and BDS-3 observations yield horizontal and vertical accuracies of 4.3 cm and 2.8 cm, respectively, comparable to CNES result, while the BDS-3-only solution performs better than CNES but is still slightly inferior to the CODE. The ZTD estimation accuracy reaches 1.8 cm for GPS+BDS-3 combinations and 2.3 cm for BDS-3-only cases. Overall, PPP-B2b delivers centimeter-level performance in real-time positioning and ZTD estimation, demonstrating strong potential as an independent, space-based precise service, though further improvement is required for GPS-only applications.]"

Comments2:

  • In lines 81-87, the authors wrote “To comprehensively evaluate the performance of PPP-B2b corrections, orbit, satellite clock and DCB, as well as the Signal-In-Space Range Error (SISRE) were analyzed, and compared with their counterparts of the final post-processed precision products provided by the Center for Orbit Determination in Europe (CODE) and the real-time precision products provided by CNES. Experiments were also conducted to evaluate the accuracies of positioning and the tropospheric zenith delay (ZTD) estimation with GPS and BDS observables collected at six stations within China.” This reviewer recommends presenting your principal contributions in the form of short highlighting phrases that can give possibility for a potential reader clear understanding novelty of this study,

Responses 2: Thank you for this suggestion. We agree that clearly highlighting the principal contributions will significantly improve the readability and impact of our manuscript.We have outlined the key innovations of this study to facilitate a quicker understanding of its main contributions for the reader."[The principal contributions of this work are threefold. First, the evaluation includes a comprehensive analysis of the product quality for orbits, clocks, DCBs, and SISRE. It then presents a critical, comparative benchmarking against established final (CODE) and real-time (CNES) precision products. The study concludes with a practical validation of the service's performance in positioning and tropospheric zenith delay (ZTD) estimation, using observational data from six stations.]"

Comments 3: Please correct line 151 where the reference number is absent.

Responses 3: Thank you for pointing this out. We agree with this comment. We have checked and corrected all the figure reference.

Comments 4: Please correct numbering of the tables (table 1 mentioned in lines 170, 237). Please renumber tables 2, 3, 4.

Responses 4: Thank you for pointing this out. We agree with this comment. We have checked and renumber tables in the manuscript.

Comments 5: The authors should use statistical procedures to justify better performance of a product type against others in tables (tables 1: lines 170, 237; table2; table 4). Standard practice in this case can include revision of the significancy in differences for metrics using statistical significance testing (e.g., paired t-tests) being performed on products. Please provide complete statistical validation to confirm whether these variations observed are not by chance.

Responses 5: Thank you for this suggestion. The results presented in the tables are derived from multi-day RMS calculations, which effectively minimize the influence of random variations.Furthermore, our comparative analysis follows the methodology adopted in related literature.

Comments 6:This reviewer recommends introducing additional discussion section where the authors can explain some principal results obtained during statistical analysis of the experiments. In this case, the conclusions section should expose principal achievements, not detailed discussion.

Responses 6: Thank you for this valuable suggestion. We have provided an in-depth analysis of the principal experimental results.The Conclusions section has been revised to focus specifically on summarizing the main achievements of this study.

Reviewer 4 Report

Comments and Suggestions for Authors

The paper presents a complete avaluation of the correction information broadcast by the Beidou B2b signal in space. Code and clock corrections were compared with CODE final products and CNES real-time corrections. PPP results including ZTD estiomates and differential code biases were also assessed.

The paper is well written and accessible. It provides and interseting performance characterization of the PPP-B2b performance. A few minor suggestions are as follows:

Page 4, please check Eq.(5) and (6): they should be identical a part the lack of the term ‘-\delta_\tau’. In Eq. (5) the term rms (root mean square) is present and it is used instead of the sigma notation. Consider to use the same sigma notation or indicate the reason for explicitly usig rms.

Page 4, line 151. Check word referencing system - It should be Figure 1.

Page 5, Figure 1. Specify the day of the DCB correction and if there was an average on the values shown in the figure. It is suggested to use a cross instead of a dot for one of the two types of DCBs

Page 5, Figure 2. Also here: explain if these are average differences computed on which period? The differences are always positive. Maybe you were considering absolute differences?

Page 7, line 200. Typo “The second-order differencing method are used” --> “is used”
Add a reference on this method.

Page 7 and next. Please carefully check the Figure numbering and referencing. The figure at the bottom of the page is again figure 1. It should be Figure 6. There are other similar problems in the rest of the paper. See for instance first figure in the next page and references in the text (Figure 3 and 9).

Page 8, check the last paragraph before section 3.4. The text seems a repetetion of previous parts.

Page 11, typo: “From the Figure 7” -> “From Figure 7”

Page 12, referencing issue on line 313

Page 15, iGMAS: please provide acronym and references to the specific product

Author Response

Thank you very much for taking the time to review this manuscript. Please find the detailed responses below and the corresponding revisions in the re-submitted files. 

Comments 1: Page 4, please check Eq.(5) and (6): they should be identical a part the lack of the term ‘-\delta_\tau’. In Eq. (5) the term rms (root mean square) is present and it is used instead of the sigma notation. Consider to use the same sigma notation or indicate the reason for explicitly usig rms.

Responses 1: Thank you for this suggestion. We have revised the equations in accordance with the relevant literature."[In practice, the two types of SISRE are calculated as[19]]"

Comments 2: Page 4, line 151. Check word referencing system - It should be Figure 1.

Responses 2: Thank you for pointing this out. We have checked and corrected all the figure references and their numbering throughout the manuscript.

Comments 3: Page 5, Figure 1. Specify the day of the DCB correction and if there was an average on the values shown in the figure. It is suggested to use a cross instead of a dot for one of the two types of DCBs

Responses 3: Thank you for this suggestion. We agree with the comment and have revised the figure caption to explicitly specify the date of the DCB correction and clarify that the values represent a daily average.

Comments 4: Page 5, Figure 2. Also here: explain if these are average differences computed on which period? The differences are always positive. Maybe you were considering absolute differences?

Responses 4: Thank you for pointing this out.  We have revised the caption for Figure 2 to explicitly state: "The differences shown are absolute differences"

Comments 5: Page 7, line 200. Typo “The second-order differencing method are used” --> “is used”
Add a reference on this method.

Responses 5: Thank you for pointing this out.  We have corrected the sentence to "The second-order differencing method is used" and have added a citation."[3.3]"

Comments 6: Page 7 and next. Please carefully check the Figure numbering and referencing. The figure at the bottom of the page is again figure 1. It should be Figure 6. There are other similar problems in the rest of the paper. See for instance first figure in the next page and references in the text (Figure 3 and 9).

Responses 6: Thank you for pointing this out. We have reviewed the entire manuscript and corrected all identified issues, including the specific examples mentioned. All figure references in the manuscript have been verified and standardized to ensure complete consistency between the text citations and the actual figure numbers.

Comments 7: Page 8, check the last paragraph before section 3.4. The text seems a repetetion of previous parts.

Responses 7:  Thank you for this suggestion.We have revised the paragraph in question on Page 8 by removing the repetitive descriptions. The changes can be found in the last paragraph before Section 3.4 in the revised manuscript."[Therefore, the standard deviation (STD) of the precise clock offset errors in PPP-B2b is used to evaluate the accuracy of the PPP-B2b clock offset corrections. In summary, the PPP-B2b product demonstrates competitive, and for BDS-3, superior clock accuracy compared to the other products.]"

Comments 8: Page 11, typo: “From the Figure 7” -> “From Figure 7”

Responses 8: Thank you for pointing this out.  We have corrected the sentence to "From Figure 7".

Comments 9: Page 12, referencing issue on line 313

Responses 9: Thank you for pointing this out. We agree with this comment. We have checked and corrected the figure reference.

Comments 10: Page 15, iGMAS: please provide acronym and references to the specific product

Responses 10:  Thank you for this suggestion. We have now defined the acronym "iGMAS" as the International GNSS Monitoring and Assessment System upon its first occurrence in the manuscript and have added citations to the official documentation describing its data products and services."[The ZTD estimates obtained from three positioning strategies—GPS-only, BDS-3-only, and GPS+BDS-3—are compared against reference values provided by iGMAS, whose final ZTD products have an accuracy better than 4 mm with a sampling interval of 2 hours]"

Round 2

Reviewer 1 Report

Comments and Suggestions for Authors

I think that my comments have addressed. Still, I see in the text broken cross-references to Figure 6 and Figure 8. However, this issue will be amended during the copy editing.

Author Response

Thank you very much for taking the time to review this manuscript. Please find the detailed responses below and the corresponding revisions in the re-submitted files. 

Comments 1: Still, I see in the text broken cross-references to Figure 6 and Figure 8. However, this issue will be amended during the copy editing.

Responses 1: Thank you for your careful review and confirmation. We appreciate your note regarding the broken cross-references to Figures 6 and 8. We will ensure that these issues are corrected during the copy-editing stage.

Reviewer 2 Report

Comments and Suggestions for Authors

My previous comments have been addressed.

I have only minor comments, which do not require another review by this reviewer:

  • please, fix the broken references in section 3.3
  • please, revise english writing another time before publication
Comments on the Quality of English Language
  • please, revise english writing another time before publication

Author Response

Thank you very much for taking the time to review this manuscript. Please find the detailed responses below and the corresponding revisions in the re-submitted files. 

Comments 1: please, fix the broken references in section 3.3

Responses 1: Thank you for the suggestion. We appreciate your minor suggestions and will carefully fix the broken references in Section 3.3.

Comments 2: please, revise english writing another time before publication.

Responses 2: Thank you for the suggestion. We have carefully revised the English throughout the manuscript and will conduct a final language polishing before publication.

Reviewer 3 Report

Comments and Suggestions for Authors

The authors have responded the comments of this reviewer.

The authors should revise text and correct lines 151, 222, 234, 260 where there are marked “Error! Reference source not found”. 

Author Response

Thank you very much for taking the time to review this manuscript. Please find the detailed responses below and the corresponding revisions in the re-submitted files. 

Comments 1: The authors should revise text and correct lines 151, 222, 234, 260 where there are marked “Error! Reference source not found”. 

Responses 1: Thank you for the suggestion. We have carefully checked the manuscript and corrected all broken cross-references appearing at lines 151, 222, 234, and 260.